# Eupalinolide J Inhibits Cancer Metastasis by Promoting STAT3 Ubiquitin-Dependent Degradation

**DOI:** 10.3390/molecules28073143

**Published:** 2023-03-31

**Authors:** Hongtao Hu, Haoyang Bai, Lili Huang, Bo Yang, Huajun Zhao

**Affiliations:** School of Pharmaceutical Sciences, Zhejiang Chinese Medical University, Hangzhou 311402, China

**Keywords:** cancer metastasis, Eupalinolide J, *Eupatorium lindleyanum* DC., STAT3, ubiquitination

## Abstract

Eupalinolide J (EJ) is an active component from *Eupatorium lindleyanum* DC. (EL), which was reported to have good antitumor activity via STAT3 and Akt signaling pathways. In this study, we identified Eupalinolide J (EJ) as a potential anti-cancer metastatic agent by target prediction and molecular docking technique screening. Follow-up experiments demonstrated that EJ exhibited a good inhibitory effect on cancer cell metastasis both in vitro and in vivo, and could effectively reduce the expression of STAT3, MMP-2, and MMP-9 proteins in cells, while the knockdown of STAT3 could weaken the inhibitory effect of EJ on cancer cell metastasis. Further molecular biology experiments revealed that EJ promoted STAT3 ubiquitin-dependent degradation, and thus, downregulated the expression of the metastasis-related genes MMP-2 and MMP-9. In conclusion, our study revealed that EJ, a sesquiterpene lactone from EL, could act as a STAT3 degradation agent to inhibit cancer cell metastasis and is expected to be applied in cancer therapy.

## 1. Introduction

Cancer is a heterogeneous disease with highly diverse features, in which unregulated cell growth leads to abnormal proliferation. Moreover, in addition to remaining in the primary lesion or local site, cancer cells can spread to other systems to form met-astatic cancer and recur in distant organs. Cancer development and metastasis are multistep processes involving complex genetic factors and physiological microenvironment. In short, genetically unstable cancer cells from primary tumors can invade their surrounding tissues, migrate into the circulatory system or body cavities, then further colonize distant organs, and generate new metastasis sites [1]. As a major cause of poor prognosis and death in cancer patients, cancer metastasis accounts for most cancer-related deaths [2]. Despite the significant advances in the detection and treatment of cancer and various targeted therapy agents have been discovered, metastatic cancer remains a thorny problem and its overall prognosis remains poor.

Natural products, which contain a large number of potential anti-tumor components, are an excellent source for the development of novel anti-cancer drugs, and many studies have obtained anti-cancer clues from natural products [3,4]. *Eupatorium lindleyanum* DC. (EL), also called ‘Yemazhui’ from its aerial part, is a traditional Chinese medicine widely used in the treatment of cough, chronic bronchitis, lobar pneumonia, and hypertension. Researchers have identified more than one hundred compounds in EL, including triterpenes, sesquiterpenes, sesquiterpene lactones, flavonoids, acyclic diterpenoids, sterols, and alkaloids [5]. Sesquiterpene lactones are one of the most abundant components in EL, and Eupalinolides (Eups) are among the main active ingredients. Eups contain two α- and β-unsaturated carbonyl groups in their structural parent nuclei, which are considered to be key to the efficacy of these compounds [6]. Moreover, special decameric and pentameric lactone rings are included, and compounds with similar structures have been found to have an effect on the ubiquitin–proteasome system [7].

Previously, Tian et al. [8] found that F1012-2, a complex composed of the three Eups Eupalinolide I, Eupalinolide J, and Eupalinolide K, could induce cell apoptosis and cell cycle arrest (G2/M) to inhibit the proliferation of MDA-MB-231 cells, and significant inhibition of Akt and activation of the p38 signalling pathway were observed in this study. Subsequently, Yang et al. [6] identified that EJ could suppress the activation of the oncogenic transcription factor STAT3 in triple-negative breast cancer (TNBC) cells and Eupalinolide O could induce apoptosis in cancer cells [9]. Further studies found that EJ could inhibit the proliferation of TNBC cells by blocking the STAT3 signaling pathway to induce apoptosis and increase the mitochondrial membrane potential and cycle arrest (G2/M) [10].

In summary, previous studies found that some Eups can significantly inhibit cancer cell proliferation by inhibiting STAT3 and Akt signaling pathways, inducing apoptosis, cell cycle arrest, and DNA damage response [6,8,9,10]. While STAT3 and Akt signaling pathways play a key role in promoting cancer metastasis [11,12,13], there are few studies and applications on the effects of the active ingredients of EL on cancer metastasis [5]. Therefore, we hope to discover the natural small molecules of Eupalinolide class that can inhibit cancer metastasis and provide help for the application of EL to cancer therapy.

In this study, we screened Eupalinolides by target prediction and molecular docking techniques and found that EJ was able to bind to a variety of metastasis-related targets and may have anti-metastatic activities. Follow-up pharmacodynamic experiments demonstrated that EJ significantly inhibited metastasis of cancer cells both in vitro and in vivo, and that the protein levels of STAT3, MMP-2, and MMP-9 could be dose-dependently reduced by EJ. Furthermore, molecular biology results revealed that EJ promoted STAT3 ubiquitin-dependent degradation and downregulated the expression of the metastasis-related genes MMP-2 and MMP-9.

## 2. Results and Discussion

### 2.1. Screening of Eupalinolides for Cancer Metastasis Inhibition

Eleven species of Eupalinolides (Eups) are present within the above-ground parts of EL, namely Eupalinolide A-E, G-K, and O [5] (Figure 1A). Nine of them are included in Pubchem. Our previous study found that Eups could affect STAT3 and AKT signaling pathways in cells, both of which were thought to play important roles in cancer metastasis.

In recent years, an increasing number of researchers have used molecular docking technology and drug screening to study drug–protein interactions. For example, Ding et al. investigated the role of GPX4 with a Parthenolide derivative in breast cancer [14], Park et al. showed that crizotinib may inhibit the kinase activity of TGFβ receptor I in metastatic lung cancer [15]. To further determine which of the above components had stronger activity against cancer metastasis, we searched the WikiPathways database [16] for proteins associated with cancers metastasis in STAT3 and AKT signaling pathways (Figure 1B) and analyzed the binding ability of Eups to these protein targets by molecular docking (Figure 1C). The network analysis identified EJ as the component of Eups most closely related to the target (Figure 1D), implying that EJ has potential anti-tumor metastatic effects.

### 2.2. EJ inhibits Metastasis of Cancer Cells In Vitro

Since glioblastoma U251 cells and breast adenocarcinoma MDA-MB-231 cells have a strong metastatic ability, these two cell lines are widely used in research on cancer metastasis. Zhu et al. studied metastasis of glioblastoma with U251 [17], and Yang et al. investigated the repression activity of coenzyme Q0 on the progression of metastasis of breast cancer with MDA-MB-231 [18]. These two cell lines were selected for in vitro experiments. We examined the effects of different concentrations of EJ (Figure 2A) on the viability of U251 and MDA-MB-231 cells for 24 h. The results showed that EJ below 5 μM was not significantly cytotoxic to these two cell lines (Figure 2B). Furthermore, the effects of non-toxic doses of EJ on cell transfer ability were examined by wound-healing (Figure 2C), transwell migration, and invasion assays (Figure 2D). The results demonstrate that EJ at safe doses can inhibit cell metastasis in a dose-dependent manner. The following experiments were then performed at concentrations of 0, 1.25, 2.5 μM.

### 2.3. EJ Inhibits Metastasis of Cancer Cells In Vivo

The effect of EJ on cancer cell metastasis in vivo was further evaluated by lung metastasis model analyses. During the experiment, there was no significant difference in body weight between the EJ-treated group and the control group of nude mice (Figure 3A). The tail vein injection of MDA-MB-231-Luc cells was found to form metastatic foci in the lungs of nude mice by in vivo imaging (Figure 3B). After surgical lung extraction, the fluorescence was observed using a live imaging system and the extent of metastasis was assessed by total fluorescence intensity in the lung (Figure 3C,D). The above results indicated that EJ could effectively inhibit the metastasis of cancer cells in vivo at a dose without significant toxicity.

### 2.4. EJ Supresses Tumor Metastasis by Inhibiting STAT3 Signaling Pathway

Previous results showed that EJ was effective in inhibiting cancer cell metastasis both in vitro and in vivo. To investigate the mechanisms of the anti-tumor metastasis of EJ, we examined the effect of EJ treatment for 12 h on metastasis-related proteins in U251 cells by Western blotting in combination with virtual screening results and found that EJ reduced the protein levels of p-STAT3, STAT3, MMP-2, and MMP-9, but had no significant effect on PI3K and AKT (Figure 4A). Therefore, we suggest that EJ may inhibit cancer cell metastasis mainly by affecting the STAT3 pathway. A significant decrease in STAT3 pathway-related proteins also occurred in EJ-treated MDA-MB-231 cells (Figure 4A). In addition, a decrease in the secreted MMP-2 and MMP-9 protein activity occurred in both cell lines after 12 h of EJ treatment (Figure 4B).

The STAT3 signaling pathway is tightly regulated to maintain a transient active state under normal physiological conditions. However, sustained STAT3 activation frequently occurs in a wide range (~70%) of human solid and hematologic malignancies [19,20,21]. Mechanistically, aberrantly activated STAT3 can promote cancer invasion and metastasis by inducing the expression of matrix metalloproteinases (MMPs, particularly MMP-1, MMP-2, and MMP-9) and other STAT3 target genes [22]. Targeting STAT3 proteins is a potentially promising strategy for tumor therapy [23]. Some of these STAT3 inhibitors have successfully entered clinical trials [24,25,26], but there is not a marketed drug.

To demonstrate that EJ exerts anti-metastatic activity through the STAT3 signaling pathway, we transfected U251 cells with STAT3-shRNA (shRNA-1 and shRNA-2) plasmids or negative control shRNA plasmids (NC-shRNA). As shown in Figure 4C, shRNA-2 transfection significantly reduced the expression level of the STAT3 protein in both cell lines. Compared to the NC-shRNA group, shRNA-2 transfection inhibited cell metastasis at 24 h. More importantly, the silencing of STAT3 by shRNA-2 significantly attenuated the ability of EJ to resist cancer cell migration compared to the NC-shRNA group (Figure 4D). These results suggest that EJ inhibits cancer cell metastasis by targeting the STAT3 pathway.

### 2.5. EJ Promotes STAT3 Protein Degradation

Since we demonstrated in our previous study that EJ inhibits cancer cell metastasis by targeting the STAT3 pathway, we next determined the effect of EJ on mRNA levels of STAT3 pathway-related genes in cancer cells by RT-PCR. Twelve hours of EJ treatment resulted in a decrease in intracellular MMP-2 and MMP-9 mRNA levels (Figure 5A). Intriguingly, the STAT3 mRNA did not change significantly after EJ treatment in U251 cells, while STAT3 mRNA levels increased in MDA-MB-231 cells (Figure 5B). It has been found that STAT3 can promote the expression of MMP-2 and MMP-9. We also found experimentally that activation of STAT3 induced upregulation of mRNA levels of these two genes in cells (Appendix A in Appendix A). The results of CHX Chase assays showed a significant decrease in the half-life of the STAT3 protein in both cell lines treated with EJ, indicating that EJ could promote the degradation of the STAT3 protein (Figure 5C,D).

During the study, we also found that, although EJ could downregulate the expression of the STAT3 protein in MDA-MB-231 and U251 cells, the trend of the STAT3 mRNA changes in the two cells were different. This may be due to the presence of feedback regulation in MDA-MB-231 cells, where drug resistance may occur over a long period of time. This suggests that our single agent targeting STAT3 may not be capable of completely treating cancer.

### 2.6. EJ Promotes STAT3 Ubiquitination

Compounds with a similar structure to EJ were found to have effects on the ubiquitin—proteasome system. After pretreatment with the proteasome inhibitor MG132, we treated both cells with EJ, and found that the EJ-induced STAT3 degradation was reversed by MG132 (Figure 6A,B).

These results suggested that STAT3 acted through the ubiquitin—proteasome system in the EJ-treated cancer cells. Ubiquitination of STAT3 was analyzed under EJ treatment. The results indicated that EJ could enhance the ubiquitination of STAT3 (Figure 6C). The results of molecular docking showed that EJ was able to bind to the DNA binding domain (DBD) of STAT3, which may also explain the ability of EJ to repress the transcription of genes downstream of STAT3 (Figure 6D). These findings suggest that EJ targets the STAT3 pathway by promoting the degradation of STAT3 ubiquitination and impeding STAT3 binding to DNA.

We all know that α- and β-unsaturated carbonyl units provide the basis for the reaction of the Michael receptor with nucleophilic reagents that may be involved in a wide range of biological activities. Various small molecules containing α and β-unsaturated carbonyl units have been found to bind to STAT3 for effective inhibition, such as HO-3867 [27], fungal metabolite galiellalactone [28], and isoalantolactone [29]. From the structural point of view, EJ contains two α, β-unsaturated carbon units and special decameric and pentameric lactone rings in its parent nucleus. Similar compounds such as parthenolide [7] and DMOCPTL [30] exhibit the effect of the intracellular proteasome pathway. Perhaps, the superposition of the two factors gives EJ the ability to promote STAT3 ubiquitination. However, the occurrence of direct binding of EJ to STAT3 and the contributing amino acid residues remain to be further investigated.

## 3. Methods and Materials

### 3.1. Molecular Docking Technology and Drug Screening

The 3D ligand structures of the Eups molecules were downloaded from PubChem (https://pubchem.ncbi.nlm.nih.gov, accessed on 12 March 2022). The receptor protein coded by the selected gene was searched in the UniProt database (https://www.uniprot.org/, accessed on 12 March 2022). We downloaded 3D structures of the proteins AKT2 (PDB ID: 1O6K), AKT3 (PDB ID: 2X18), JNK3 (PDB ID: 2P33), JUN (PDB ID: 5T01), MMP2 (PDB ID: 3AYU), MMP8 (PDB ID: 1A85), PAK1 (PDB ID: 1YHV), PIK3CD (PDB ID: 6PYR), RAC1 (PDB ID: 1MH1), STAT3 (PDB ID: 6QHD), and TRIO (PDB ID: 1NTY) from RCSB PDB (https://www.rcsb.org/, accessed on 12 March 2022) and EFNA1 (Uniprot ID: P20827)and PIK3CB (Uniprot ID: P42338) from the AlphaFold Protein Structure Database. All chains of the structures were used to prepare the receptor, and all water molecules around the protein were removed. AutoDock software was used to carry out hydrogenation and charge calculation of the proteins. Finally, 10 docking poses were obtained for the molecule by AutoDock Vina (v 1.2.2) [31], and the one with the optimal binding energy was chosen. Visualization of docking used R-4.1.3 and Pymol.-2.5 and network analysis used CytoScape (v3.9.1) to filter potential drugs based on degree.

### 3.2. Cell Culture and Reagents

MDA-MB-231 and U251 cell lines were obtained from the Chinese Academy of Sciences. Cells were maintained in DMEM (Gibco, Waltham WLM, MA, USA) containing 10% fetal bovine serum (FBS, Sijiqing, Hangzhou, China) at 37 °C with 5% CO_2_ in an incubator.

EJ was isolated from the *Eupatorium lindleyanum* DC. herb from our co-lab as previously described [5]. Antibodies against STAT3 (#30835), p-STAT3 (#4113), AKT (#4691), PI3K (#4249S), MMP-2 (#4022), MMP-9 (#13667), JAK1 (#3344), and GAPDH (#5174) were obtained from Cell Signaling Technology.

### 3.3. MTT Assay

The inhibitory effects of EJ on the growth of cancer cells were evaluated by MTT assay. Cells (5 × 10^3^ cells/well) were planted on a 96-well plate at 37 °C for 24 h before treatment. After that, different dosages of EJ were subjected to incubation with cancer cells for 48 h. After incubation, 20 μL of MTT reagent was added to each well and the plates were incubated for an additional 4 h. After that, the formazan crystals were dissolved in DMSO, and the absorbance was measured at 550 nm using a microplate reader.

### 3.4. Wound-Healing Assay

Cells were seeded in a 24-well plate and allowed to form a confluent monolayer. Subsequently, the monolayers were scratched by 200 μL pipette tips to form perpendicular wounds. The scratched well was washed three times with PBS and fresh medium was added in each well. The scratched areas were photographed under a microscope (LH-M100CB-1, Nikon) at appropriate time points. The degree of wound closure was calculated as the percentage of the area covered by the cells relative to the area at 0 h and analyzed by Image J 1.52a to measure the width of the wound.

### 3.5. Transwell Migration and Invasion Assay

A 6.5 mm pore size chamber (Corning, Corning, NY, USA) was used to study cell migration and invasion. Cells were placed on the top surface of the upper chamber of polycarbonate permeable filters for permeable migration assay or polycarbonate permeable filters coated with 1 μg Matrigel for permeable invasion assay. The chambers were incubated at 37 °C in a humidified atmosphere of 5% CO_2_ for 12–24 h. After incubation, the cells that migrated or invaded the lower side of the filter were fixed with methanol and stained with crystal violet. Images of five random areas were captured from each membrane using a microscope.

### 3.6. RT-PCR

Total RNA was isolated from the sample using TRIZOL reagent (Invitrogen, Waltham WLM, MA, USA). cDNA was synthesized using PrimeScript™ RT kit (Takara Bio, Inc., Kusatsu shi, Japan) following the manufacturer’s protocol. qPCR was performed in a CFX96 Real-Time system (Bio-Rad Laboratories, Inc., Hercules HERC, CA, USA) with SYBR-Green (Bio-Rad Laboratories, Inc.) as the fluorescent dye. Primers were designed to target the gene of interest. The PCR reaction was performed with the following conditions: 95 ℃ for 10 min, followed by 40 cycles of 95 °C for 10 s and 60 °C for 30 s. A melting curve analysis was performed to ensure the specificity of the PCR product. The relative expression levels were calculated using the 2^−ΔΔCq^ method.

Primer sequences are shown below.

Human STAT3 forward: 5′-CAGCAGCTTGACACACGGTA-3′ and reverse: 5′-AAACACCAAAGTGGCATGTGA- 3′;

Human MMP-2 forward: 5′-TACAGGATCATTGGCTACACACC-3′ and reverse: 5′-GGTCACATCGCTCCAGACT- 3′;

Human MMP-9 forward: 5′-TGTACCGCTATGGTTACACTCG-3′ and reverse: 5′-GGCAGGGACAGTTGCTTCT- 3′.

### 3.7. Western Blotting Analysis

Cells were lysed on ice with RIPA lysis buffer containing PMSF and aprotinin. The concentrations of total protein were measured by BCA assay. Equal amounts of protein were then separated on SDS-PAGE gels. The proteins were transferred to polyvinylidenedifluoride (PVDF) membranes (Millipore). The membranes were blocked with 5% skimmed milk for 2 h at room temperature, followed by incubation with primary antibodies overnight at 4 °C. On the next day, the membranes were washed with TBST (Tris-Buffered Saline Tween-20) 5 times at 25 min intervals, and then incubated with appropriate secondary antibodies at room temperature for 2 h. Finally, the membranes were washed with TBST again and chemiluminescence detection was performed by ECL (Bio-Rad, USA).

### 3.8. Gel Zymography

Gel Zymography is a technique used for the analysis of proteolytic activity of enzymes. This technique involves the separation of proteins by electrophoresis on a polyacrylamide gel containing a proteolytic substrate. The samples were subjected to SDS-PAGE electrophoresis in a 9% acrylamide-bis-gel (Millipore-Sigma, St. Louis, MO, USA) containing 0.1% gelatin. After washing, the gels were incubated in an activation buffer (50 mmol/L Tris-HCl, 6 mmol/L CaCl2, 1.5 μmol/L ZnCl_2_, pH 7.4) containing 2.5% Triton X-100 for 1 h, followed by 24 h at 37 °C, followed by staining with Coomassie brilliant blue R-250 (Bio-Rad, Hercules, CA, USA) and immersion in a decontamination solution (40% methanol, 10% acetic acid, 50% water). The gels were then digitized and densitometric analysis was carried out using Image J 1.52a.

### 3.9. ShRNA Design and Transfection

ShRNAs for STAT3 were designed by Genechem (Shanghai, China). The targeted target sequences were as follows:

STAT3-shRNA#1 241-ACAATCTACGAAGAATCAA-2553;

STAT3-shRNA#2 241-GGCAACAGATTGCCTGCATT-2553.

The transfection of shRNA into cells was performed with Lipofectamine 2000. Lipofectamine 2000 was diluted 25 times in reduced serum medium Opti-MEM I (1×) (Gibco, USA), and STAT3-shRNA or negative control plasmids were diluted to equal volume. After 5 min of incubation, diluted Lipofectamine 2000 and STAT3-shRNA or negative control plasmids were mixed and incubated for 20 min. Subsequently, MDA-MB-231 and U251 were cultured in Opti-MEM I (1×) containing the above mixture, respectively. After 4 to 6 h, the medium mixture was replaced with a fresh DMEM medium.

### 3.10. Lung Metastasis Model

The animal studies were approved by the Institutional Animal Care and Use Committee of Zhejiang Chinese Medical University (No. 20210615-11) on 15 June 2022, and performed in accordance with the Guidelines proposed by the Laboratory Animal Research Center of Zhejiang Chinese Medical University.

MDA-MB-231-Luc cells (5 × 10^5^, in 0.1 mL DMEM/F12 serum-free medium) were injected intravenously through the tail vein into 4-week-old BALB/c nu/nu female mice. Mice were randomly assigned to vehicle control and EJ (20 mg/kg). After injection, the mice were treated with EJ (30 mg/kg) or saline once every 2 days for 18 days. Mice were then anesthetized and subjected to in vivo imaging system (Nikon) imaging. After sacrifice, the lungs were taken, and the fluorescence intensity was observed by a live imaging system to analyze the lung metastasis of the tumor.

### 3.11. Statistical Analysis

All data were analyzed using GraphPad Prism 5.0 statistical software (San Diego, CA, USA). Results were conducted in at least three independent experiments and are presented as mean ± SD. Statistical analysis was performed using a two-tailed Student’s *t*-test. The criteria for statistical significance were * *p* < 0.05; ** *p* < 0.01; *** *p* < 0.001.

## 4. Conclusions

A sesquiterpene lactone component EJ was selected from Traditional Chinese Medicine EL, which has anti-tumor metastasis effects both in vitro and in vivo and no obvious toxicity at the therapeutic concentrations in both environments. The anti-metastatic effect of EJ is STAT3-dependent. EJ could inhibit the function of STAT3 by binding to its DNA binding domain or promote STAT3 ubiquitination and degradation to reduce the STAT3 content in cancer cells. Activation of STAT3 could significantly increase the expression of the metastasis-related proteins MMP-2 and MMP-9 in cancer cells. Treatment with EJ inhibits STAT3, which reduces the expression and secretion of the MMP-2 and MMP-9 proteins and exerted an anti-tumor metastasis effect.

In conclusion, we found that an active ingredient EJ in EL could act as a STAT3 degrader to inhibit cancer cell metastasis and would be a promising agent for cancer therapy.

## Figures and Tables

**Figure 1 molecules-28-03143-f001:**
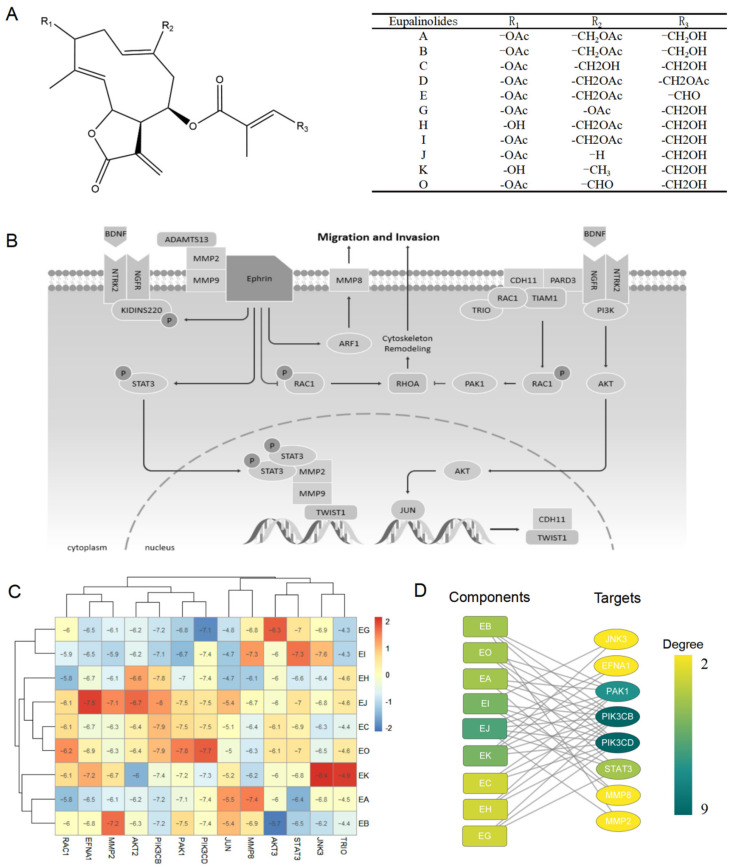
Eups screening for anti-cancer metastases. (**A**) The chemical structure of Eupalinolides (Eups). (**B**) STAT3 and AKT signaling pathways associated with cancer metastasis. (**C**) Heat map of Eups docking binding to metastasis-associated STAT3 and AKT signaling pathways proteins. (**D**) Analysis of the binding network of Eups to STAT3 and AKT signaling pathway-related proteins.

**Figure 2 molecules-28-03143-f002:**
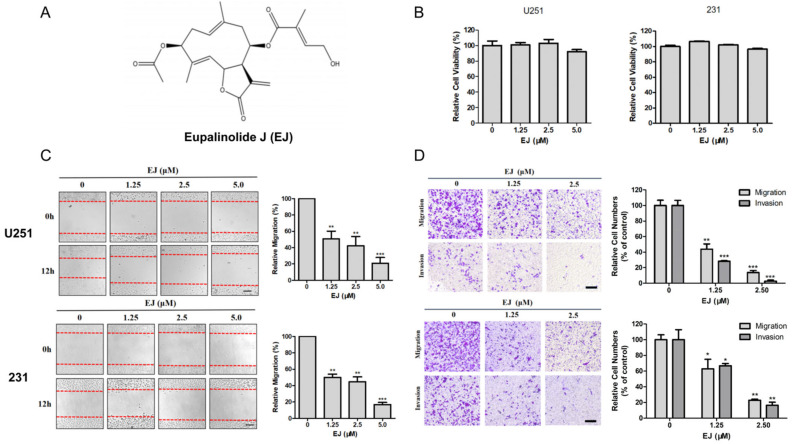
EJ inhibited the metastasis of cancer cells in vitro. (**A**) The chemical structure of Eupalinolide J (EJ). (**B**) Non-significant cytotoxic doses of EJ were determined on U251 and MDA-MB-231 cells by MTT assay. (**C**) EJ treatment with non-toxic doses significantly reduced wound closure rate of cancer cells in wound-healing assay. (**D**) EJ significantly inhibited cancer cell migration and invasion by transwell assay (* *p* < 0.05, ** *p* < 0.01, *** *p* < 0.001).

**Figure 3 molecules-28-03143-f003:**
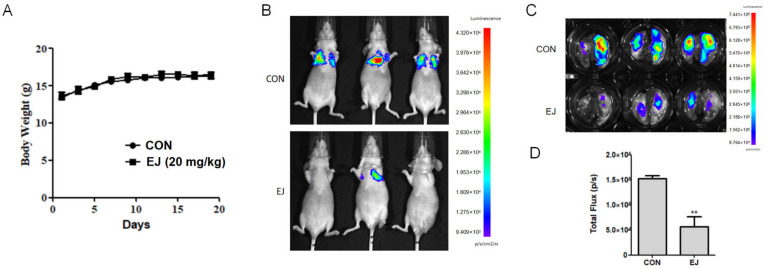
EJ inhibited the metastasis of cancer cells in vivo. (**A**) EJ treatment had no significant effect on the body weight of nude mice. (**B**) Inhibitory effect of EJ on cancer cell metastasis in nude mice was evaluated by in vivo imaging. (**C**) Anatomical lung extraction and in vivo imaging of cancer cell metastasis in the lung. (**D**) The total fluorescence intensity of lung cancer cells was significantly decreased in the EJ-treated group compared to the control group (** *p* < 0.01).

**Figure 4 molecules-28-03143-f004:**
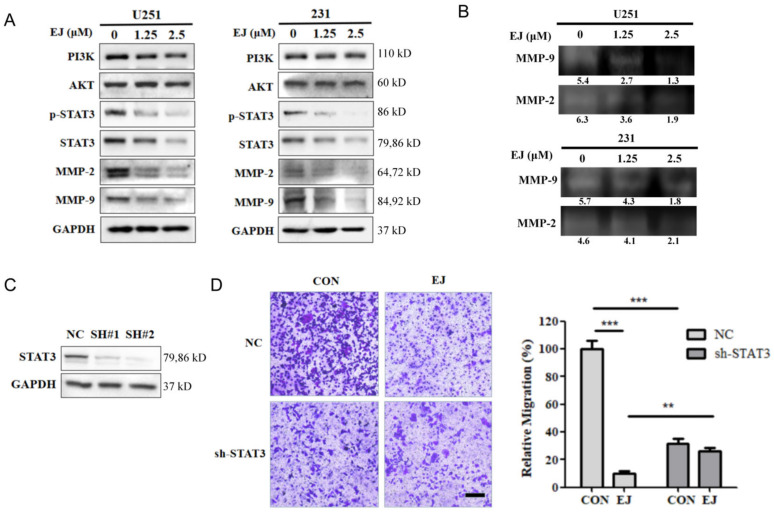
EJ suppressed cancer cell metastasis through the STAT3 signaling pathway. (**A**) Effect of EJ on the expression level of STAT3 and AKT signaling pathway-related proteins were detected by Western blot. (**B**) EJs inhibitory effect on the activity of MMP-2 and MMP-9 was detected by gelatinase profiling assay (the gray-scale quantification values are reported below the strips). (**C**) Knockdown of STAT3 in U251 cells by shRNA transfection. U251 cells were transfected with STAT3-shRNA (SH#1 and SH#2) plasmids or negative control shRNA plasmids (NC). The expression of STAT3 in U251 cells was determined using Western blotting. (**D**) Knockdown of STAT3 significantly inhibited the migration of U251 cells and reversed the ability of EJ to inhibit metastasis in different groups of cells using transwell assays (** *p* < 0.01, *** *p* < 0.001).

**Figure 5 molecules-28-03143-f005:**
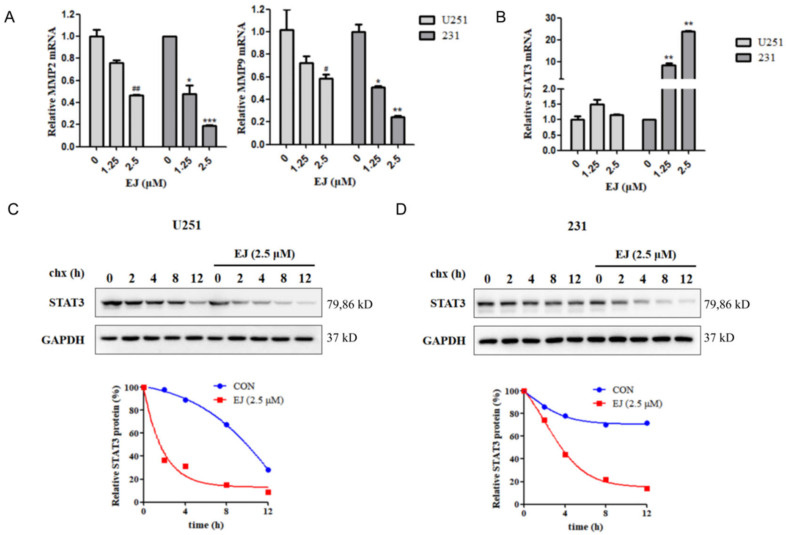
EJ promoted the degradation of STAT3 protein. (**A**) EJ dose-dependently downregulates MMP-2 and MMP-9 mRNA levels in cancer cells. (**B**) Effect of EJ on the expression of the STAT3 mRNA in cancer cells. EJ had no significant effect on the STAT3 mRNA in U251 cells, but upregulated the expression of the STAT3 mRNA in MDA-MB-231 cells. (^#^ vs. U251 control: ^#^
*p* < 0.05, ^##^
*p* < 0.01. * vs. 231 control: * *p* < 0.05, ** *p* < 0.01, *** *p* < 0.001). (**C**,**D**) EJ reduced the half-life of the STAT3 protein in U251 (**C**) and MDA-MB-231 (**D**) cells, as shown by CHX chase assay.

**Figure 6 molecules-28-03143-f006:**
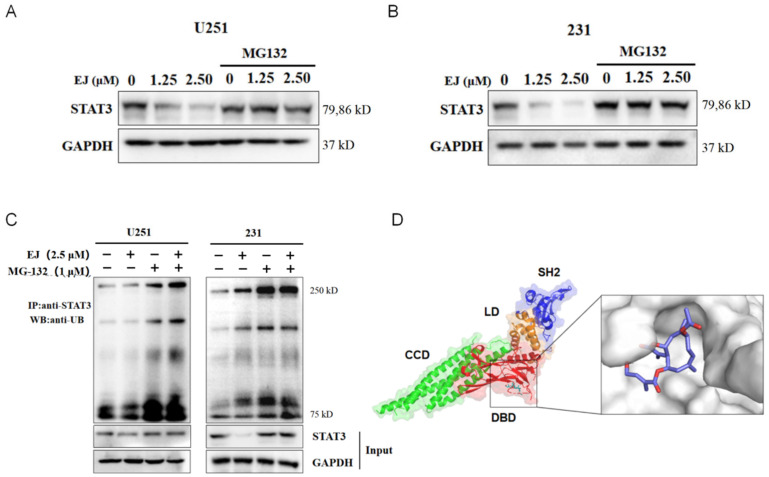
EJ promoted STAT3 ubiquitin-dependent degradation. (**A**,**B**) The ubiquitin–proteasome inhibitor MG132 reversed the inhibitory effect of EJ on the expression of the STAT3 protein in U251 (**A**) and MDA-MB-231 (**B**) cells. (**C**) EJ was found to promote STAT3 protein ubiquitination modifications by immunoprecipitation. (**D**) Molecular docking results showed that EJ was able to bind to the DNA binding domain of the STAT3 protein (PDB ID: 6QHD). (CCD: coiled-coil domain, DBD: DNA binding domain; LD: linker domain; SH2: Src homology 2 domain).

## Data Availability

The datasets used and/or analyzed during the current study available from the corresponding author on reasonable request.

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
