# Peer review of "Eupalinolide J Inhibits Cancer Metastasis by Promoting STAT3 Ubiquitin-Dependent Degradation"

_molecules, 2023, doi:10.3390/molecules28073143_

Round 1

Reviewer 1 Report

Response to Title: Eupalinolide J inhibits cancer metastasis by promoting STAT3 ubiquitin-dependent degradation

This study examines the effects of Eupalinolide J (EJ) is an active component from Eupatorium lindleyanum DC. inhibits cell metastasis by promoting STAT3 in vitro and in vivo. Results of this study can be applied to use of EJ to be applied in cancer therapy. The methods employed could answer the objective, result and discussion is clear. However, little mistakes should be addressed as follows:

·        All Figure had a low solution. Please change to high solution

Author Response

Point 1: All Figure had a low solution. Please change to high solution.

Response 1: Thank you for your suggestion.We have replaced them with higher resolution pictures in the revised version.

Reviewer 2 Report

Journal of molecules

Research Article;

The article entitled Eupalinolide J inhibits cancer metastasis by promoting STAT3 ubiquitin-dependent degradation’’. The authors best explain the Eupalinolide J antitumor activity via STAT3 and Akt signaling pathways. the author identified Eupalinolide J (EJ) as a potential anti-cancer metastatic agent by target prediction and molecular docking technique screening. Results demonstrated that EJ had good inhibitory effect on cancer cell metastasis both in vitro and in vivo, and could effectively reduce the expression of STAT3, MMP-2, MMP-9 protein in cells, while knockdown of STAT3 could weaken the inhibitory effect of EJ on cancer cell metastasis. the study revealed that EJ, a sesquiterpene lactone from EL, could act as a STAT3 degradation agent to impedes cancer cell metastasis and is expected to be applied in cancer therapy.  

I carefully read the manuscript and found it suitable for publication in the journal. I accept this article for possible publication. There are some minor mistakes in the article which should be corrected by the authors. After the correction of all the mistakes and revision, the article could be considered for publication in the prestigious molecules Journal.

Comments for Authors

Ø  Write keywords in alphabetical order.

Ø  Section Introduction; Revise it. The authors needs to include more refferance and make it more explinatory.

Ø  Use chemdraw to draw the structure. Why The structure of Eupalinolides repeated in the figure 1 and figure 2.  

Ø  The needs to use original image size so that the picture looks more clear.

Ø  Thr author needs to write the conclusion of the study.

Ø  Use EndNote or Mendeley software for reference sequences.

Ø  Check grammatically and spelling throughout the manuscript. There are some mistakes.

Ø  Write the protein kDa in the figure.

Cite the following references;

v  doi.org/10.2174/1871520622666220831124321

v  doi.org/10.1038/s41419-021-03771-z

Author Response

Point 1: Write keywords in alphabetical order.

Response 1: Thank you for your suggestion.We have modified the keyword ranking.

Point 2: Section Introduction; Revise it. The authors needs to include more refferance and make it more explinatory.

Response 2: Thank you for your suggestion.We expanded the introduction and cited relevant literature (37 lines).

Point 3: Use chemdraw to draw the structure. Why The structure of Eupalinolides repeated in the figure 1 and figure 2.  

Response 3: We used chemraw to draw the structure. Figure 2 shows the structure of EJ. This compound is the main research object of subsequent research, so it is shown in detail in Figure 2.

Point 4: The needs to use original image size so that the picture looks more clear.

Response 4: Thank you  for your suggestion. In the revised version, we replaced all the pictures with the original ones.

Point 5: The author needs to write the conclusion of the study.

Response 5: Thank you for your suggestion.We have added conclusion to the revised version.

Point 6:  Use EndNote or Mendeley software for reference sequences.

Response 6: Thank you for your suggestion.We used EndNote software for reference sequences.

Point 7: Check grammatically and spelling throughout the manuscript. There are some mistakes.

Response 7: Thanks for your suggestion. We have tried our best to polish the language in the revised manuscript.

Point 8: Write the protein kDa in the figure.

Response 8: Thank you for your suggestion.We annotated the molecular weight of each protein next to it.

Reviewer 3 Report

Hongtao Hu and co work is focused on the effect of Eupalinolide J (EJ) on metastatic cancer cells. They have chosen two different models and have also performed an analysis on the in vivo effect of this compound. Experiments are globally well conducted and convincing. Their results bring interesting data since they found that EJ strongly reduce migration, invasion as well as metastatic formation in mice. However, few points should be clarified. 

The authors should specify the administration mode of mice with the compound? Is-it every day?  

The results of MMP2 activity (zymmography) are not clear enough and quantification should be done for such experiments. Moreover, the authors should explored in parallel the activity of MMP9.

Experiments performed, in cells in which STAT3 expression were inhibited, indicate that this protein could be involved in the anti-metastatic effect of EJ. An overexpression of STAT3 could be interesting to confirm these conclusions. 

Author Response

Point 1: The authors should specify the administration mode of mice with the compound? Is-it every day?  

Response 1: Thank you for your suggestion.We administered the drug every two days and added this information to the 3.10 Lung metastasis model.

Point 2: The results of MMP2 activity (zymmography) are not clear enough and quantification should be done for such experiments. Moreover, the authors should explored in parallel the activity of MMP9.

Response 2: Thank you for your suggestion. We carried out experiments to detect the activity of MMP9, used image J for quantitative analysis, and modified figure 4 B.

Point 3: Experiments performed, in cells in which STAT3 expression were inhibited, indicate that this protein could be involved in the anti-metastatic effect of EJ. An overexpression of STAT3 could be interesting to confirm these conclusions. 

Response 3: We agree that more studies would be useful to understand the details of interaction.  However, our experimental results show that STAT3 knockdown can significantly reverse the anti-metastasis effect of EJ, which can support our conclusion, and many studies have used similar methods[1-2]. We will study further in the future.

  • Pore N, Wu S, Standifer N, et al. Resistance to Durvalumab and Durvalumab plus Tremelimumab Is Associated with Functional STK11 Mutations in Patients with Non-Small Cell Lung Cancer and Is Reversed by STAT3 Knockdown. Cancer Discov. 2021;11(11):2828-2845. doi:10.1158/2159-8290.CD-20-1543
  • Yang L, Zhou F, Zhuang Y, et al. Acetyl-bufalin shows potent efficacy against non-small-cell lung cancer by targeting the CDK9/STAT3 signalling pathway. Br J Cancer. 2021;124(3):645-657. doi:10.1038/s41416-020-01135-6

Round 2

Reviewer 3 Report

The authors answered all my remarks and added the requested experiments so I approve the publication of this work